# Effect of Input C/N Ratio on Bacterial Community of Water Biofloc and Shrimp Gut in a Commercial Zero-Exchange System with Intensive Production of *Penaeus vannamei*

**DOI:** 10.3390/microorganisms10051060

**Published:** 2022-05-20

**Authors:** Wujie Xu, Guoliang Wen, Haochang Su, Yu Xu, Xiaojuan Hu, Yucheng Cao

**Affiliations:** 1Shenzhen Base of South China Sea Fisheries Research Institute, Chinese Academy of Fishery Sciences, Shenzhen 518121, China; xu_wujie@163.com (W.X.); su.haochang@163.com (H.S.); xuyublq@163.com (Y.X.); xinr129@163.com (X.H.); 2South China Sea Fisheries Research Institute, Chinese Academy of Fishery Sciences, Guangzhou 510300, China; guowen66@163.com; 3Key Laboratory of South China Sea Fishery Resources Exploitation & Utilization, Ministry of Agriculture and Rural Affairs, Guangzhou 510300, China

**Keywords:** biofloc system, *Penaeus vannamei*, C/N ratio, bacterial community, gut microbiota, nitrogen dynamics, production performance

## Abstract

Although increasing attention has been attracted to the study and application of biofloc technology (BFT) in aquaculture, few details have been reported about the bacterial community of biofloc and its manipulation strategy for commercial shrimp production. An 8-week trial was conducted to investigate the effects of three input C/N ratios (8:1, 12:1 and 16:1) on the bacterial community of water biofloc and shrimp gut in a commercial BFT tank system with intensive aquaculture of *P. vannamei*. Each C/N ratio group had three randomly assigned replicate tanks (culture water volume of 30 m^3^), and each tank was stocked with juvenile shrimp at a density of 300 shrimp m^−3^. The tank systems were operated with zero-water exchange, pH maintenance and biofloc control. During the trial, the microbial biomass and bacterial density of water biofloc showed similar variation trends, with no significant difference under respective biofloc control measures for the three C/N ratio groups. Significant changes were found in the alpha diversity, composition and relative abundance of bacterial communities across the stages of the trial, and they showed differences in water biofloc and shrimp gut among the three C/N ratio groups. Meanwhile, high similarity could be found in the composition of the bacterial community between water biofloc and shrimp gut. Additionally, nitrogen dynamics in culture water showed some differences while shrimp performance showed no significant difference among the three C/N ratio groups. Together, these results confirm that the manipulation of input C/N ratio could affect the bacterial community of both water biofloc and shrimp gut in the environment of a commercial BFT system with intensive production of *P. vannamei*. Moreover, there should be different operations for the nitrogen dynamics and biofloc management during shrimp production process under different C/N ratios.

## 1. Introduction

The Pacific white shrimp, *Penaeus vannamei*, is the dominant economic crustacean species in global seafood aquaculture. The aquaculture production of this species almost doubled during the past decade, reaching a global production of 5.4 million tons and an economic value of 32.1 billion dollars in 2019 [1]. Biofloc technology (BFT) should be one of the most important technical innovations that contributes to the sustainable development of shrimp aquaculture in the past decade [2,3,4]. At present, increasing research on BFT is still widely carried out to improve its practical application in intensive aquaculture of *P. vannamei* worldwide [4,5,6].

The BFT is essentially a microbial manipulation approach applied in shrimp aquaculture systems. The basis of BFT is active suspended bioflocs, which are microbial aggregates mainly consisting of dense bacterial communities in the water column of aquaculture systems [7,8,9,10]. Along with the formation and development of bioflocs in the culture water, they can assimilate and transfer toxic nitrogenous compounds, such as ammonium nitrogen (NH_4_^+^-N) and nitrite nitrogen (NO_2_^−^-N), thereby maintaining a suitable water quality to sustain safe and intensive culture of shrimp under limited water exchange operations [5,7,11]. Meanwhile, the bioflocs generated in the culture water can be ingested and utilized in situ by cultured shrimp, supplementing natural nutrition and enhancing healthy growth for the shrimp [4,5,12]. In practice, the establishment, development and management of mature biofloc in situ culture water is the core and key task for successful production of shrimp in biofloc-based intensive aquaculture systems [5,13,14].

Inoculating mature biofloc-rich water is an effective operation to establish a biofloc-based system for shrimp intensive culture [5,15]. After the establishment of a biofloc-based system, there is no need to add organic carbon into culture water for the growth of heterotrophic bacteria and the development of biofloc. This is different from the traditional proposition, which highlights the increase in input C/N ratio through organic carbon addition to accelerate the assimilation of NH_4_^+^-N by heterotrophic bacteria and convert it to bacterial protein without producing NO_2_^−^-N and nitrate nitrogen (NO_3_^−^-N) [16,17,18]. However, continuous addition of organic carbon can generate large amounts of heterotrophic biofloc biomass in the culture system, which increases not only the input cost but also the management difficulties in terms of solids removal and oxygen supplement [5,19,20]. More importantly, increasing the C/N ratio can induce a shift of the bacterial community of biofloc [10,15,19], which could have very significant impacts on water quality control and shrimp production performance [15,21]. It is thought that the input C/N ratio is one of the critical factors affecting growth rate of different microbial communities, thereby generating different nitrogen conversion pathways and microbial biomass yields [15,18,19].

In order to advance the efficient utilization of biofloc, there requires accurate and thorough understanding of its bacterial communities for the manipulation of microbial structure and function through input C/N ratio. Increasing research has recently investigated the bacterial community of the bioflocs from shrimp aquaculture systems using 16S rRNA gene sequencing analysis [21,22,23,24,25]; these studies were conducted in small-scale laboratory systems under controlled conditions. The study of Panigrahi et al. [21] showed that increasing the input C/N ratio could significantly change the diversity and community structure of the bacterial communities from the bioflocs in the culture water. We have attempted to explore the abundance, diversity and composition of bacterial communities in the bioflocs from a commercial BFT system with shrimp intensive production [14], and in this study we further investigate their responses to different input C/N ratios under zero-water exchange.

Gut microbiota play important roles in maintaining growth and health of cultured shrimp. Extensive studies have shown that the rearing environments could be a potential and important source of gut microbiota of the shrimp [26,27]. The biofloc systems are found to be one of the most abundant and diverse microbial-based aquaculture systems so far, which is characterized by dense microbial aggregates suspended in the aquaculture water [10]. The bacterial communities existing in bioflocs could inevitably have the potential to influence the gut microbiota of cultured shrimp, which could subsequently affect their physiological processes, welfare and growth [12,28,29,30,31]. Furthermore, the shrimp can ingest the biofloc continually from the culture water, which could further strengthen the connection of the bacterial community between water biofloc and shrimp gut [30,31]. However, it is not clear whether shrimp gut microbiota would change with the shifting of the bacterial community of biofloc responding to different C/N ratios.

In the context above, this study aims to investigate the effects of different input C/N ratios on the diversity, composition and abundance of bacterial communities in water biofloc and shrimp gut from a commercial BFT system with intensive production of *P. vannamei* under zero-water exchange. Since the bacterial communities of bioflocs are main drivers for inorganic nitrogen control and ecological nutrition cycling in the culture systems, the nitrogen dynamics and shrimp performance responding to different C/N ratios are also discussed to provide guidance for process management during shrimp production in BFT systems.

## 2. Materials and Methods

### 2.1. Trial Shrimp and System Preparation

The study was conducted at the Shrimp Aquaculture Station facilities (N 22°43′31″, E 115°35′2″) of South China Sea Fisheries Research Institute, located in the city of Shanwei, Guangdong province, China. Natural seawater was pumped from Red Bay of Shanwei, and then chlorinated before use in this study. Ten-day-old postlarvae (PL_10_) of *P. vannamei* were obtained from Hainan Haishang Shrimp Breeding Co., Ltd., (Wenchang, Hainan, China). Postlarvae were stocked into a 72 m^2^ biofloc-based nursery pond at a density of 3200 shrimp m^−3^, and reared for 35 days under limited water exchange. During the nursery period, molasses was added to the culture water daily to achieve an input C/N ratio of 12:1 based on the carbon–nitrogen contents of the applied feed and the carbon content of the molasses [15]. The main water quality characteristics of the nursery pond on day 35 are provided in Appendix A.

After the 35-day nursery period, nine tank systems were prepared in a greenhouse with semitranslucent plastic shed for the study. The tank system was designed and constructed for shrimp intensive culture at commercial scale. Each tank system consisted of a concrete tank (6 m/length × 6 m/width × 1 m/height, water volume 30 m^3^), nine water injectors (Yangjiang Shrimp Bio-Tech Co., Ltd., Yangjiang, China), a 750 W circulating pump (SBP100, Guangdong Lingxiao Pump Industry Co., Ltd., Yangjiang, China) and a small-size foam fractionator (working volume of 60 L) (Figure 1). Thirty cubic meters of water, half of which was biofloc-rich water from the same nursery pond and the other half sand-filtered seawater, was pumped into each tank. Meanwhile, juvenile shrimp (2.68 ± 0.44 g) from the nursery pond were trapped using a nylon cage net, group weighed and immediately stocked into the trial tanks to reach a density of 300 shrimp m^−3^ water volume.

### 2.2. Trial Design and Culture Management

Three input C/N ratios of 8:1, 12:1 and 16:1 were established in this study, namely, groups CN8, CN12 and CN16, respectively. The C/N ratio was based on the calculation of the carbon–nitrogen contents of the applied feed and the supplemented molasses [15]. Each group had three randomly assigned tank systems. The feed was a commercial pellet feed (40.0% protein, 8.0% lipid, 3.5% fiber and 14% ash) and purchased from Guangdong Guangxin Feed Co., Ltd. (Maoming, Guangdong, China). The feed had a calculated C/N ratio of 8:1; no supplementation of molasses was designated as the control group (CN8). The molasses (34% *w*/*w* carbon and specific gravity of 1.4) was purchased from a sugarcane plant. For groups CN12 and CN16, 0.57 and 1.10 mL of molasses were supplemented daily for every 1 g of the feed offered, respectively.

The shrimp were fed the feeds by the automatic feeders at twelve times in equal portions daily. Details of feed management and molasses supplementation were described in our previous study [14]. As needed, the foam fractionator was operated to control biofloc volume between 10 and 15 mL L^−1^ and volatile suspended solids (VSS) between 150 and 300 mg L^−1^ in the culture water of each tank system [11]. Throughout the trial, no water was exchanged in all tanks; as needed, freshwater was added into tanks for offsetting losses to evaporation and waste discharge. Sodium carbonate (Na_2_CO_3_, purity > 99.5%) was added as needed to maintain pH above 7.0 and meanwhile compensated for alkalinity loss. The operation time of the foam fractionator and amount of sodium carbonate added were recorded. The shrimp were cultured for eight weeks in the trial.

### 2.3. Water Quality Monitoring and Shrimp Performance Determination

Every day at 10:00, the salinity, temperature, dissolved oxygen (DO) concentration and pH of tank water were measured using a handheld YSI-650 multiparameter meter (Yellow Springs Instruments Inc., Yellow Springs, OH, USA). Biofloc volume (BFV) was measured daily on-site with Imhoff cones, registering the volume taken in by the settable flocs in 1000 mL of culture water after 30 min of settling. Every week, culture water was collected from each tank to analyze alkalinity, TAN, NO_2_^–^-N, nitrate-nitrogen (NO_3_^–^-N), total nitrogen (TN) and VSS following *Standard Methods for the Examination of Water and Wastewater* [32]. VSS is used to quantitatively estimate the microbial biomass and the development of biofloc [15].

After eight weeks, the water of each tank was drained successively. Shrimp were harvested with dip nets into baskets, and then weighed to obtain total harvest biomass of each tank. Meanwhile, two hundred shrimp from each tank were randomly chosen for the determination of harvest weight. The selected indicators of production performance were calculated using the following equations:growth rate (g week^−1^) = shrimp harvest weight/culture weeks,(1)
survival rate (%) = 100 × shrimp harvest biomass/shrimp harvest weight/shrimp stocking count,(2)
yield (kg m^−3^) = shrimp harvest biomass/tank water volume,(3)
feed conversion ratio (FCR) = total offered feed weight/shrimp harvest biomass.(4)

### 2.4. Microbial Sampling, DNA Extraction and qPCR Analysis

Culture water containing biofloc was collected from each tank by sterile bottles for microbial analysis on day 28 (middle stage) and day 56 (late stage) of the trial. The inoculated water containing biofloc from nursery pond was also collected as the initial stage sample (day 0). Collected water (200 mL) was immediately filtered through a 0.2 μm pore-size polycarbonate membrane filter (Millipore Corporation, Billerica, MA, USA) to obtain a biofloc sample. Microbial DNA was extracted from biofloc samples using the E.Z.N.A.^®^ Soil DNA Kit (Omega Bio-tek, Norcross, GA, USA) according to manufacturer’s protocols.

The 16S rRNA genes of extracted DNA for bacteria were quantified by SYBR green real-time quantitative polymerase chain reaction (qPCR) using FTC-3000^TM^ real-time PCR systems (Funglyn Biotech Inc., Toronto, ON, Canada). The primer was designed as 515F (GTGCCAGCMGCCGCGGTAA)/926R (CCGTCAATTCMTTTGAGTTT) [33]. The procedures of PCR amplification and qPCR reaction followed our previous study [34]. Reported qPCR reactions had amplification efficiencies > 94% and R^2^ > 0.99. The gene copy number was calculated by cycle threshold (Ct) value and the standard curve. All the standard DNA and test samples were run in triplicate. The total bacterial density of water biofloc was estimated as 16S rRNA gene copies per milliliter of water volume.

### 2.5. HiSeq Sequencing and Data Processing

The 16S rRNA genes of extracted DNA for bacteria were further investigated by HiSeq sequencing. The extracted DNA was amplified on V4-V5 regions using primers of 515F 5′-barcode- GTGCCAGCMGCCGCGG)-3′ and 907R 5′-CCGTCAATTCMTTTRAGTTT-3′. PCR reactions were performed in triplicate 20 μL mixture containing 4 μL of 5 × FastPfu Buffer, 2 μL of 2.5 mM dNTPs, 0.8 μL of each primer (5 μM), 0.4 μL of FastPfu Polymerase and 10 ng of template DNA. The amplification products were purified using the AxyPrep DNA Gel Extraction Kit (Axygen Biosciences, Union City, CA, USA) and then sequenced on an Illumina HiSeq platform (PE250, Illumina, San Diego, CA, USA) at Hangzhou Mingke Bio Co., Ltd. (Hangzhou, China). The raw reads were deposited into the NCBI Sequence Read Archive (SRA) database (Accession No. PRJNA817401).

Raw fastq files were demultiplexed, quality-filtered using QIIME (version 1.17) following the pipelines of quality control [35]. Operational Units (OTUs) were clustered with 97% similarity cutoff using UPARSE (version 7.1) (http://drive5.com/uparse/, accessed on 28 October 2020), while chimeric sequences were identified and removed with UCHIME. Taxonomic identity of each representative sequence was determined using the RDP Classifier (http://rdp.cme.msu.edu/, accessed on 28 October 2020) against the SILVA (SSU123) 16S rRNA database using a confidence threshold of 70%. Species richness estimator of Chao1 and Shannon diversity index was calculated based on Mothur v.1.21.1. Community evenness was calculated as Pielou’s evenness [36]:Community evenness = H/log (S),(5)
where H is the Shannon diversity index and S is the number of species. Principal coordinates analysis (PCoA) was performed to examine dissimilarity in the bacterial community structure based on unweighted Unifrac distances matrix in QIIME 2 (https://qiime2.org/, accessed on 28 October 2020).

### 2.6. Statistical Analysis

All statistical analyses were performed using IBM SPSS Statistics 20.0 software for Windows (IBM Corporation, Armonk, NY, USA). Data of water quality parameters (salinity, temperature, dissolved oxygen, pH, alkalinity, TAN, NO_2_^−^-N, NO_3_^−^-N, BFV and VSS), and bacterial density, alpha diversity indexes and relative abundance of community were firstly analyzed using two-way ANOVA with time and C/N ratio as fixed factors. When C/N ratio had a significant effect, one-way ANOVA was used to test for significant difference between treatments at individual time points. Data of shrimp productive indexes (final weight, growth rate, survival, yield and FCR) were analyzed using one-way ANOVA. All data were subjected to homogeneity of variance test before ANOVA. Differences were considered significant at *p* < 0.05. When a significant difference was found, Tukey’s test was used to identify difference between groups.

## 3. Results

### 3.1. Water Quality and Shrimp Performance in Intensive BFT Systems

All selected water quality parameters were maintained within acceptable ranges for *P. vannamei* culture throughout the 8-week trial (Appendix A). The results of shrimp performance indicators are presented in Table 1. No significant differences were found in the harvest weight, growth rate, survival rate, yield and FCR among the three groups (*p* > 0.05). As the C/N ratio increased from 8:1 to 16:1, the harvest weight and growth rate of shrimp showed an increase tendency while the survival rate showed a decrease tendency.

### 3.2. Effect of C/N Ratio on Nitrogen Dynamics and Biofloc Management

The changes in TAN, NO_2_^–^-N, NO_3_^–^-N and TN concentrations are shown in Figure 2. TAN and NO_2_^–^-N concentrations fluctuated significantly (*p* < 0.05) among most sampling times, and both of them remained at low levels throughout the 8-week trial in the three C/N ratio groups, not exceeding 1.23 and 1.26 mg L^−1^, respectively (Figure 2A,B). NO_3_^–^-N and TN concentrations increased significantly (*p* < 0.05) over time during the 8-week trial, and both of them showed fluctuations at some sampling times (Figure 2C,D). At most sampling times, NO_3_^–^-N and TN concentrations decreased as the C/N ratio increased from 8:1 to 16:1 (Figure 2C,D).

There were differences in the operations of culture management among the three C/N ratio groups. The operation time of the foam fractionator and additional amount of sodium carbonate increased significantly as the C/N ratio increased from 8:1 to 16:1 (Table 2).

Quantitative change in water biofloc is shown in Figure 3. The microbial biomass of water biofloc increased significantly (*p* < 0.05) within the first two weeks and then fluctuated between 150 and 300 mg L^−1^ of VSS until the end of the trial; there were no significant differences (*p* > 0.05) at most sampling times among the three C/N ratio groups (Figure 3A). The bacterial density of water biofloc increased significantly (*p* < 0.05) during the initial to middle stages and then remained around 2 × 10^12^ copies L^−1^ of 16S rRNA gene during the middle to later stages; there was no significant difference (*p* > 0.05) among the three C/N ratio groups (Figure 3B). There were significant positive correlations (*p* < 0.05) between mg L^−1^ of VSS and copies L^−1^ of 16S rRNA gene for all three C/N ratios (Figure 3C). The bacterial density of water biofloc increased as its microbial biomass increased.

### 3.3. Effect of C/N Ratio on Bacterial Community in Water Biofloc and Shrimp Gut

Alpha diversity of the bacterial community in water biofloc and shrimp gut is shown in Figure 4. In water biofloc, Chao1 richness, Shannon diversity and community evenness increased highly significantly (*p* < 0.01) during the initial to middle stages, and then remained at relatively stable levels during the middle to later stages in the three C/N ratio groups. No significant difference (*p* > 0.05) was found in Chao1 richness among the three C/N ratio groups, and significant decreases (*p* < 0.05) in Shannon diversity and community evenness were found as the C/N ratio increased from 8:1 to 16:1. In shrimp gut, Chao1 richness increased highly significantly (*p* < 0.01) during the initial stage to the middle and later stages in the three C/N ratio groups, while Shannon diversity and community evenness deceased highly significantly (*p* < 0.01) during the initial stage to the middle and later stages in the C/N ratios of 8 and 16. Significant differences (*p* < 0.05) were found in Chao1 richness, Shannon diversity and community evenness at the middle stage among the three C/N ratio groups, with the highest Chao1 richness for the C/N ratio of 8 and the highest Shannon diversity and community evenness for the C/N ratio of 12.

PCoA analysis showed that the structure of the bacterial community could be obviously separated by culture stages rather than C/N ratios in both water biofloc and shrimp gut (Figure 5). For main bacterial genus in water biofloc, relative abundances of *Vibrio*, *NS9 marine group_norank*, *Cryomorphaceae_uncultured*, *Marinomonas*, *Owenweeksia* and *Tenacibaculum* decreased significantly (*p* < 0.05), while *Tropicibacter*, *Saprospiraceae_uncultured*, *Planctomyces*, *Rhodobacteraceae_uncultured*, *Pir4 lineage*, *Ardenticatenia_norank*, *Microbacterium* and *Candidatus Alysiosphaera* increased significantly (*p* < 0.05) during the initial stage to middle and later stages in the three C/N ratio groups (Figure 6A). Significant differences were found in relative abundances of *Rhodobacteraceae_uncultured*, *Candidatus Alysiosphaera* and *NS9 marine group_norank* at both the middle and later stages among the three C/N ratio groups (Figure 6B). For main bacterial genus in shrimp gut, relative abundances of *Vibrio*, *Spongiimonas* and *Pseudoalteromonas* increased significantly (*p* < 0.05) while *Planctomyces*, *Gammaproteobacteria Incertae Sedis_uncultured* and *JTB255 marine benthic group_norank* decreased significantly (*p* < 0.05) during the initial stage to middle and later stages in the three C/N ratio groups (Figure 6C). Significant differences were found in relative abundances of *Vibrio and Planctomyces* at the middle stage and *Candidatus Alysiosphaera* at the later stage among the three C/N ratio groups (Figure 6D).

### 3.4. Relationship of Bacterial Community between Water Biofloc and Shrimp Gut

The numbers of shared OTUs between water biofloc and shrimp gut increased significantly (*p* < 0.05) during the initial stage to middle and later stages in the three C/N ratio groups (Figure 7A). Shared OTUs between water biofloc and shrimp gut accounted for higher than 83% and 89% of total OTUs in water biofloc at the middle and later stages, respectively, and no significant difference (*p* > 0.05) was found among the three C/N ratio groups (Figure 7A). Among several dominant bacterial species with relative abundance of more than 5%, *Vibrio_Unclassified* and *Tropicibacter naphthalenivorans* were two core-shared species existing in both water biofloc and shrimp gut (Figure 7B). Moreover, at the middle and late stages of the trial, much higher relative abundances of *Vibrio_Unclassified* were detected in shrimp gut (26.4~60.2%) than those in water biofloc (0.2~0.7%), while similar relative abundances of *Tropicibacter naphthalenivorans* were detected in both water biofloc (7.5~13.2%) and shrimp gut (4.3~7.5%).

## 4. Discussion

### 4.1. C/N Ratio Affected Insignificantly on Shrimp Production Performance in Intensive BFT System

The C/N ratio is considered to be a key factor for the operation of BFT in an intensive aquaculture system. In this study, the input C/N ratio had no significant effect on the production performance of *P. vannamei* intensively cultured in a commercial BFT system under zero-water exchange. This is similar to the results on culture performance of *P. vannamei* in biofloc-based intensive systems by C/N ratio manipulation or molasses supplementation in previously studies [11,14,37]. However, our further research also observed significant differences for growth performance and feed utilization of *P. vannamei* under different input C/N ratios (from 9 to 18) in biofloc-based zero-exchange tanks [15]. The difference should be attributed mainly to the fact that higher C/N ratio can induce more production of biofloc and thereby causes stress and suppresses growth of the shrimp [15,37,38].

Water quality maintenance and feed nutrition supply are two main aspects that together determine shrimp production performance in a biofloc-based system. In this study, all monitored water quality parameters were maintained within acceptable ranges for *P. vannamei* culture throughout the 8-week trial; there were no significant differences among the three C/N ratio groups. It is worth pointing out that the alkalinity and biofloc level were maintained within appropriate ranges in the three C/N ratio groups during the entire trial [11,37,39]. On the other hand, the high protein feed together with improved feeding management could provide sufficient nutrition for shrimp growth during the trial [11,14,40]; in this respect, there was no difference for the three C/N ratio groups. Therefore, it is not surprising to observe insignificant difference on shrimp growth and feed utilization among the three input C/N ratio groups in this study.

### 4.2. C/N Ratio Altered Nitrogen Dynamics and Culture Management in Intensive BFT System

Harmful nitrogen control is the core task for the operation of BFT in intensive aquaculture systems. In this study, both TAN and NO_2_^–^-N concentrations were maintained within acceptable ranges for shrimp intensive culture during the entire trial, and the dynamics of them showed similar change patterns under the three C/N ratios. Meanwhile, significant differences were found on the dynamics of NO_3_^–^-N and TN under the three C/N ratios, which showed a faster increase in NO_3_^–^-N and TN concentrations with the decrease in C/N ratio. Based on the characteristics of the three nitrogen conversion pathways in biofloc-based systems, we deduced that autotrophic nitrification (TAN assimilation to NO_2_^–^-N and then to NO_3_^–^-N) probably is the main TAN removal pathway in groups with C/N ratios of 8:1 and 12:1, while TAN assimilation by heterotrophic bacteria into bacterial biomass could dominate in groups with C/N ratios of 16:1 [15,19,41].

Not only nitrogen dynamics were altered by C/N ratio, but also different culture management measures needed to be applied in the three C/N ratio groups in this study. During the trial, BFV and VSS increased faster as C/N ratio increased, which indicates that higher C/N ratio favored the growth of heterotrophic bacteria and further produced more bioflocs in the zero-water-exchange systems [15,19]. Correspondingly, a longer time was needed for the operation of the foam fractionator in the groups with higher C/N ratio during the trial (Appendix A), which was necessary to control BFV between 10 and 15 mL L^−1^ and VSS between 150 and 300 mg L^−1^ of the culture water [11]. At the same time, more Na_2_CO_3_ was added in the higher C/N ratio groups to adjust for the decrease in pH in these groups resulting from the increased CO_2_ production by the higher biomass of heterotrophic bacteria [15,19,39]. The higher level of CO_3_^2−^ supplementation for pH correction in groups with C/N ratio of 12 and 16 resulted in higher alkalinity in these tanks at the end of the trial [15].

### 4.3. C/N Ratio Affected the Bacterial Community in Water Biofloc and Shrimp Gut

The change in VSS over time can reflect the microbial biomass and the development of the biofloc in the culture water [11,15]. In the present study, a significant increase in microbial biomass of water bioflocs was found during the initial stage, indicating overall fast growth of microorganisms in the culture water under zero-water exchange and continual feeding inputs. The microbial biomass of bioflocs was then maintained in an appropriate range (150~300 mg L^−1^ of VSS) by the separation of the foam fractionator as needed during the middle and later stages. This operation held enough and appropriate amounts of active microorganisms for waste nitrogen transformation, and alleviated oxygen consumption to benefit the shrimp culture [11,14,20,38,42]. The results of 16S rRNA gene copies L^−1^ in water biofloc during the initial, middle and later stages further indicated the change in total bacterial density in bioflocs as they developed during the study [43]. The significant positive correlations between mg L^−1^ of VSS and copies L^−1^ of 16S rRNA gene indicate that the bacteria should be the main component of the microbial biomass of water biofloc in this study. Overall, the microbial biomass and total bacterial density of water biofloc showed no significant differences among the three C/N ratio groups in this study.

The bacterial communities in water biofloc are responsible for nitrogen transformation processes and water quality maintenance [7,8,11,14,19]. In this study, the alpha diversity of bacterial communities in water biofloc increased significantly firstly and then remained relatively stable. These changing trends were consistent with those of total bacterial density in water biofloc across the initial, middle and later stages, indicating that there is a time process of adaptation and maturation for bacterial growth and biofloc development during the start and management of a biofloc-based intensive culture system [14,34]. Further, significant decrease was found in the Shannon diversity and community evenness of bacterial communities in water biofloc as C/N ratio increased from 8 to 16, which is probably because higher C/N ratio favored the growth of heterotrophic bacteria and prompted some specific bacterial groups to be dominant [15,44], thereby reducing the distribution uniformity of bacterial communities in water biofloc. Meanwhile, the Shannon diversity and community evenness of bacterial communities in shrimp gut showed opposite change trends compared to those in water biofloc; they had the lowest levels in the group with the C/N ratio of 16. The results were similar to those reported in recent studies on biofloc-based shrimp culture systems [23,31]. Together, these results indicate that the input C/N ratio had significant effect on the alpha diversity of bacterial communities in both water biofloc and shrimp gut in biofloc-based intensive culture systems.

Moreover, the bacterial communities in water biofloc showed obvious clusters at the initial, middle and later stages in the three C/N ratio groups, indicating that they developed to different structure status across the culture stages. This phenomenon was not obvious in bacterial communities in shrimp gut. The results indicate that the structure of bacterial communities were more stable in shrimp gut than that in water biofloc during the process of shrimp culture in intensive systems. Further analysis revealed that C/N ratio had a significant effect on the relative abundance of dominant bacterial genus in both water biofloc and shrimp gut. Previous studies have also shown that increasing input C/N ratio could change the microbial community in the culture water of biofloc-based systems and induce the accumulation of some specific bacterial groups [44,45]. For example, in the trial, CN12 and CN16 groups had significantly higher proportions of *Rhodobacteraceae_uncultured* genus in water biofloc than that in the CN8 group. The *Rhodobacter* group is considered an excellent biofilm-forming organism, and could possibly play an important role in the establishment of beneficial bacterial communities of water biofloc [29].

### 4.4. High Similarity of Bacterial Community between Water Biofloc and Shrimp Gut

In this study, most of the bacterial OTUs in shrimp gut could be found in water biofloc, indicating the high phylogenetic similarity of bacterial community composition between water biofloc and shrimp gut in the intensive BFT system. The similar bacterial community composition between water biofloc and shrimp gut have also been found in previous experimental studies [29,31,46]. In the BFT system, *P. vannamei* can ingest water biofloc in situ as a supplemental food source [4,47]. In this study, it is safe to speculate that as ingested biofloc entered into the gut of cultured shrimp, its bacterial community could influence the microbiota of shrimp gut in some way [31]. For example, certain bacterial species of ingested biofloc could have ability to survive and colonize in shrimp gut, and thereby become parts of the bacterial community in shrimp gut [29,31]. As seen in this study, *Vibrio_Unclassified* and *Tropicibacter naphthalenivorans* were two core dominant bacterial species existing in both water biofloc and shrimp gut.

Although core bacterial community showed high similarity in the composition between water biofloc and shrimp gut, they had significantly different proportions. The observed dissimilarity might be that the presence of selective pressures leads to the recruitment of specific microbial inhabitants within the shrimp gut [29]. In this study, it is worth noting that the proportion of *Vibrio* genus (detected one species of *Vibrio_Unclassified*) in bacterial communities was very low in water biofloc while it was very high in shrimp gut. *Vibrio* genus is one of the most common bacterial groups in the shrimp and surrounding environment [10,48], and they are likely to thrive under the environment of BFT systems in which shrimp were cultured [46]. Despite the presence of their pathogenic or opportunistic pathogenic strains, most of shrimp inhabitant *Vibrio* spp. were determined to be nonpathogenic [10,49]. Moreover, some were previously shown to be capable of assimilating TAN and NO_3_^–^-N accumulated in the biofloc system [10,50]. Unfortunately, the *Vibrio* spp. detected were unclassified in this study, which makes it difficult to determine the nature of impact caused by its high abundance in the bacterial communities within shrimp gut.

## 5. Conclusions

This study confirms that the manipulation of input C/N ratio could affect the bacterial community of both water biofloc and shrimp gut in the environment of a commercial BFT system with intensive production of *P. vannamei*. During the trial, the differences in nitrogen dynamics and culture management measures of the tank systems probably resulted from the responses of the bacterial community in water bioflocs to different C/N ratios. Furthermore, no significant effect could be found in the shrimp production performance by C/N ratio. Together, these results suggest that the input C/N ratio should be an operation strategy mainly for nitrogen transformation control rather than shrimp performance improvement in production practice. In this respect, the nitrogen-cycling microbiomes of biofloc need to be identified and characterized in-depth for BFT aquaculture systems. The nutrient recycling and nitrogen budget of the system also needs to be assessed. These further studies will not only establish the microbial foundation for nitrogen transformation and nutrient recycling in situ water of the BFT system, but will also help to manipulate functional biofloc for efficient nitrogen management in intensive shrimp production.

## Figures and Tables

**Figure 1 microorganisms-10-01060-f001:**
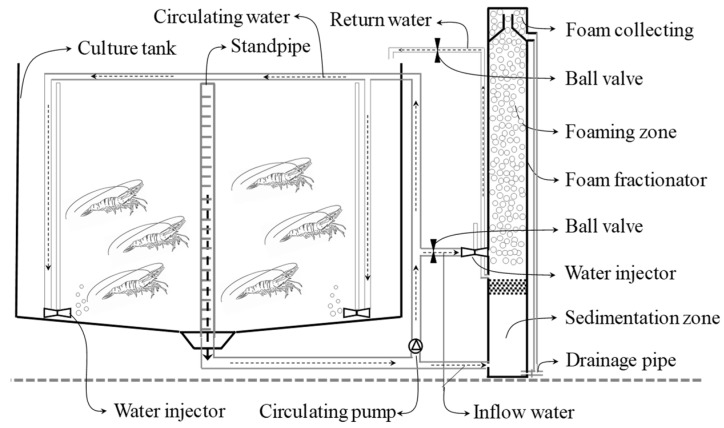
Schematic diagram of a commercial biofloc technology system with zero-water exchange for shrimp intensive production. Arrows indicate water flow. Water was pumped from the culture tank by running the circulating pump; most of water circulated to the culture tank through water injectors. The remaining water flowed into a foam fractionator and then returned to the culture tank. The water entering and leaving the foam fractionator was adjusted by ball valves at water flow rates of 15~25 L min^−1^.

**Figure 2 microorganisms-10-01060-f002:**
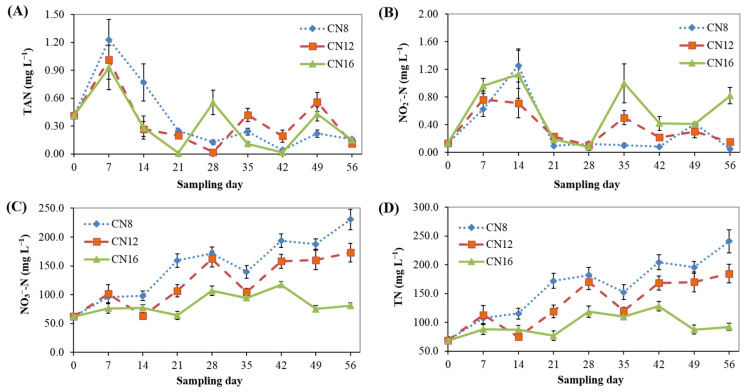
Nitrogen dynamics in the culture water of biofloc-based tank systems during an 8-week intensive production trial of *P. vannamei* with three input C/N ratios (means ± S.D., n = 3): (**A**) TAN, total ammonia nitrogen; (**B**) NO_2_^−^-N, nitrite nitrogen; (**C**) NO_3_^−^-N, nitrate nitrogen; (**D**) TN, total nitrogen.

**Figure 3 microorganisms-10-01060-f003:**
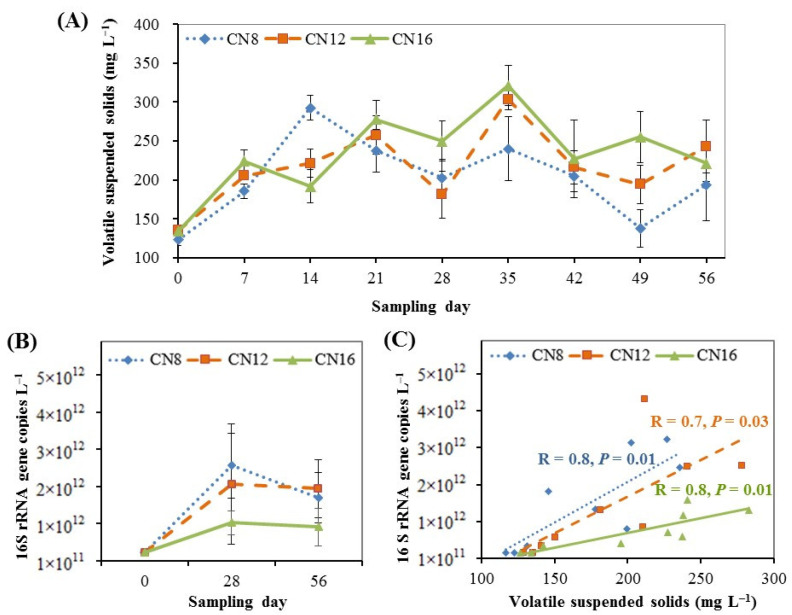
Changes and relations of microbial biomass and bacterial density in water biofloc from biofloc-based tank systems during an 8-week intensive production trial of *P. vannamei* with three input C/N ratios (means ± S.D., n = 3): (**A**) Change of microbial biomass of water biofloc; (**B**) Change of bacterial density of water biofloc; (**C**) Linear correlation between microbial biomass and bacterial density of water biofloc.

**Figure 4 microorganisms-10-01060-f004:**
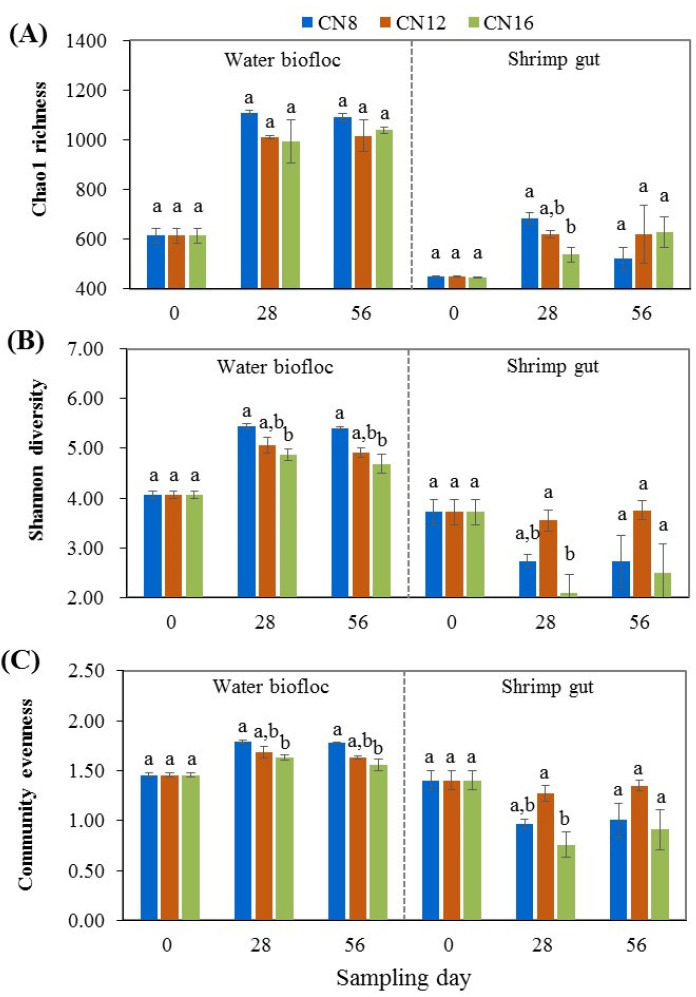
Alpha diversity of the bacterial community in water biofloc and shrimp gut from biofloc-based tank systems during an 8-week intensive production trial of *P. vannamei* with three input C/N ratios (means ± S.D., n = 3). Different letters indicate significant difference at the same time point among the three C/N ratio groups.

**Figure 5 microorganisms-10-01060-f005:**
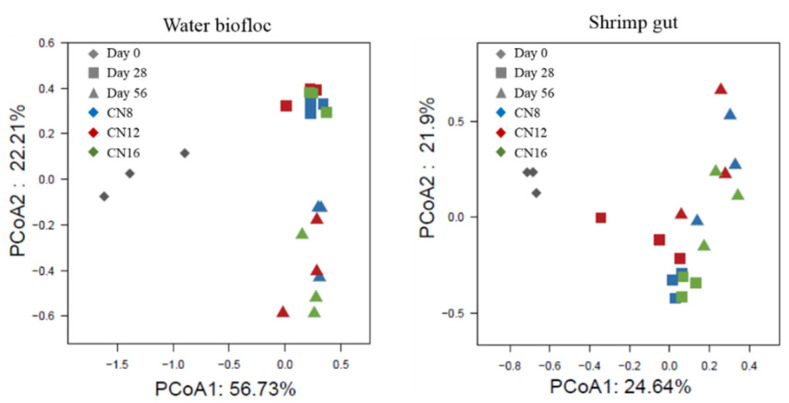
The differences in bacterial community structure in water biofloc and shrimp gut from biofloc-based tank systems during an 8-week intensive production trial of *P. vannamei* with three input C/N ratios, identified by principal coordinates analysis (n = 3).

**Figure 6 microorganisms-10-01060-f006:**
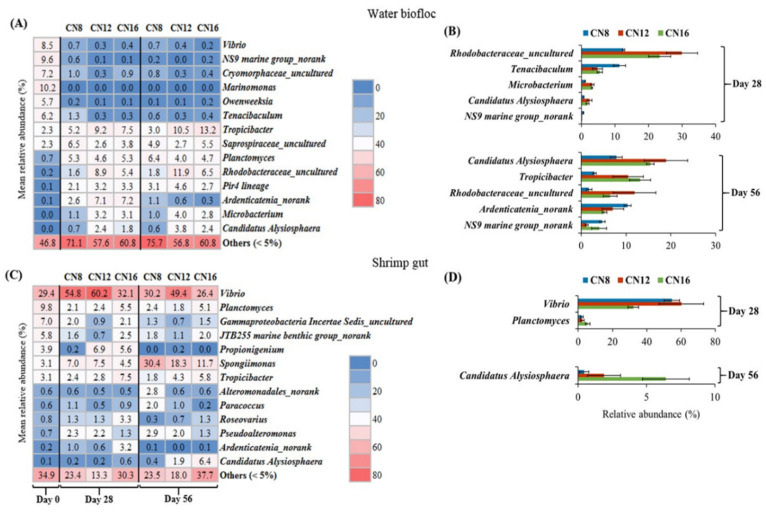
Taxonomy composition of the bacterial community at genus level in water biofloc and shrimp gut from biofloc-based tank systems during an 8-week intensive production trial of *P. vannamei* with three input C/N ratios (means ± S.D., n = 3). (**A**) Changes in main bacterial genera in water biofloc and their differences among the three C/N ratios. (**B**) Bacterial genera in water biofloc with significant differences in relative abundance among the three C/N ratios. (**C**) Changes in main bacterial genera in shrimp gut and their differences among the three C/N ratios. (**D**) Bacterial genera in shrimp gut with significant differences in relative abundance among the three C/N ratios.

**Figure 7 microorganisms-10-01060-f007:**
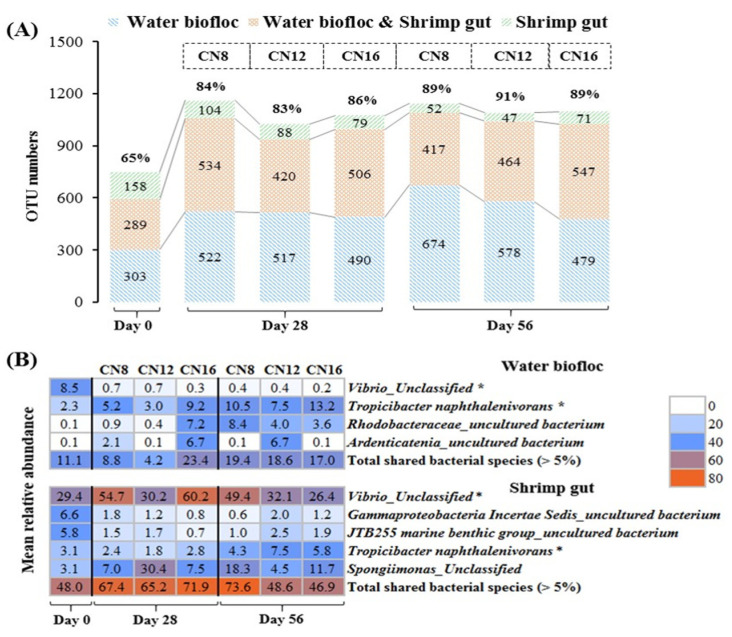
OTUs similarity and core bacterial species between water biofloc and shrimp gut from biofloc-based tank systems during an 8-week intensive production trial of *P. vannamei* with three input C/N ratios (means ± S.D., n = 3). (**A**) The number and similarity of shared and unique bacterial OTUs between water biofloc and shrimp gut. (**B**) The difference in core bacterial species in relative abundance between water biofloc and shrimp gut. Significant difference among the three C/N ratio groups indicated by *.

**Table 1 microorganisms-10-01060-t001:** Production performance of *P. vannamei* in biofloc-based tank systems during an 8-week intensive production trial with three input C/N ratios (means ± S.D., n = 3).

Group	Harvest Weight (g)	Growth Rate (g wk^−1^)	Survival Rate (%)	Yield (kg m^−3^)	Feed Conversion Ratio
CN8	15.93 ± 0.30	1.66 ± 0.04	92.8 ± 2.7	4.43 ± 0.18	1.43 ± 0.03
CN12	16.31 ± 0.60	1.70 ± 0.07	92.4 ± 3.1	4.52 ± 0.15	1.38 ± 0.01
CN16	16.79 ± 1.10	1.76 ± 0.14	89.7 ± 3.0	4.51 ± 0.15	1.39 ± 0.03

**Table 2 microorganisms-10-01060-t002:** The culture management for biofloc-based tank systems during an 8-week intensive production trial of *P. vannamei* with three input C/N ratios (means ± S.D., n = 3).

Group	CN8	CN12	CN16
Foam fractionator operation time (h)	336 ± 15 ^a^	409 ± 27 ^b^	510 ± 31 ^c^
Sodium carbonate addition amount (kg)	26 ± 3 ^a^	29 ± 4 ^a^	39 ± 6 ^b^

Different superscript letters indicate significant difference among the three C/N ratio groups.

## Data Availability

The raw read datasets for this research can be found in the NCBI Sequence Read Archive database (Accession No. PRJNA817401) (https://www.ncbi.nlm.nih.gov/sra/, accessed on 19 March 2022).

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
