# Peer review of "Effect of Input C/N Ratio on Bacterial Community of Water Biofloc and Shrimp Gut in a Commercial Zero-Exchange System with Intensive Production of Penaeus vannamei"

_microorganisms, 2022, doi:10.3390/microorganisms10051060_

Round 1

Reviewer 1 Report

Author Xu et al. describes "Changes in Bacterial Community of Water Biofloc and Shrimp Gut, and Their Responses to Input C/N ratio during Intensive Production of Penaeus vannamei in a Biofloc-based System">

This paper can be accepted after considering the following comments;

1: The title is confusing

2: Is there also assimilation of some toxic metals such as Pb, Cd, Hg, Cr or any other toxic organic pollutant by P. vannamei during the experiment

3:Line 181-182, if there is no significant difference I do not think it is good to present Table 1 in the main text, you can sift to the supplementary file. You can show the Figure S1 as a main figure in the main text instead as a supllementry file.

4: Resolution of Figure -3 need to improve.

5: Provide some more future prespectives.

Good luck

Author Response

The paper deals with an interesting topic of biofloc formation and management in "zero exchange" white shrimp culture. It is well written but in poor English language that needs to be improved in spite of the fact that the sentences are understandable.

>> Thank you for your comments. We have tried again to improve language editing, please see revised MS.

“The study aimed to investigate the effects of different input C/N ratios on the diversity, composition and abundance of bacterial communities in water biofloc and shrimp gut from P. vannamei intensive production systems under practical environmental conditions”. Most of these effects are already known from the works of various authors and even from an earlier article by the same authors. The production system used is closer to the commercial system and this was the main reason for this research. The author used the term "practical environmental conditions" to describe the conditions in the production tanks (eg lines 17, 71, etc.), which does not really mean or describe anything. It needs to be changed.

>> Thank you for your good comments. We have rephrased related terms and sentences. Please see lines 16-17, 27-31, 99-101, 120-122, 642-643.

In general, Abstract, Introductiom, Materials and Methods, Results and Discussion are correctly presented, and described in details.

>> Thanks.

In the summary the results of the MS confirm (CONFIRM BUT DON'T BRING SOMETHING COMPLETELY NEW!) the findings obtained under laboratory conditions in the environment of a commercial white shrimp production. This shoud be emphasized in the final statement in the Conclussion of the MS.

>> Thank you for your good suggestion. We have rephrased the finding and importance of this study in the part of Conclusions; please see revised MS.

Lines 70-72 should be moved in the paragraph that explains the aim of the paper.

>> Completed, see lines 120-122 in revised MS.

English should be revised. It is understandable, but not correctly written.

>> Thanks. We have tried again to improve language editing, please see revised MS.

Reviewer 2 Report

The paper deals with an interesting topic of bioflock formation and management in "zero exchange" white shrimp culture. It is well written but in poor English language that needs to be improved in spite of the fact that the sentences are understandable.

“The study aimed to investigate the effects of different input C/N ratios on the diversity, composition and abundance of bacterial communities in water biofloc and shrimp gut from P. vannamei intensive production systems under practical environmental conditions”. Most of these effects are already known from the works of various authors and even from an earlier article by the same authors. The production system used is closer to the commercial system and this was the main reason for this research. The author used the term "practical environmental conditions" to describe the conditions in the production tanks (eg lines 17, 71, etc.), which does not really mean or describe anything. It needs to be changed.

In general, Abstract, Introductiom, Materials and Methods, Results and Discussion are correctly presented, and described in details.

In the summary the results of the MS confirm (CONFIRM BUT DON'T BRING SOMETHING COMPLETELY NEW!) the findings obtained under laboratory conditions in the environment of a commercial white shrimop production. This shoud be emphasized in the final statement in the Conclussion of the MS.

Lines 70-72 should be moved in the paragraph that explains the aim of the paper.

English should be revised. It is understandable, but not correctly written.

Author Response

Responses to reviewer 2:

This paper can be accepted after considering the following comments;

1: The title is confusing.

>> Thanks. We have rewrite the title.

2: Is there also assimilation of some toxic metals such as Pb, Cd, Hg, Cr or any other toxic organic pollutant by P. vannamei during the experiment.

>> Thanks for your good question. There exists the potential risk of heavy metals accumulation in zero-exchange BFT shrimp production systems. The concern has been raised by previous study (Prangnell et al., Some Limiting Factors in Superintensive Production of Juvenile Pacific White Shrimp, Litopenaeus vannamei, in No-water-exchange, Biofloc-dominated Systems. Journal of the World Aquaculture Society 2016, 47(3):396-413); and in the trial of this study the measured heavy metal concentrations in shrimp tail muscle were all within acceptable limits for human consumption (CEFAS 1998; FDA 2000, 2011). We have also detected the concentrations of some heavy metals (e.g. Fe, Cu, Mn, Mo, Cr) in both culture water and shrimp body in our previous study (data not published); and there were not obvious accumulation of any metals for a whole cycle of shrimp production. Normally, if the input feed meets safety and quality requirements for shrimp production, there should not be any toxic pollutants accumulation in the shrimp body. On the contrary, the shrimp that were produced in the BFT system with zero-water exchange probably be safer without the entrance of any exogenous harmful substances.

3: Line 181-182, if there is no significant difference I do not think it is good to present Table 1 in the main text, you can sift to the supplementary file. You can show the Figure S1 as a main figure in the main text instead as a supplementary file.

>> Thanks for your good suggestions. We have presented Table 1 as a supplementary file (Table S2) and showed Figure S1 in the main text (Figure 1). Please see the revised MS.

4: Resolution of Figure -3 need to improve.

>> Thanks. We have improved the resolution of Figure 3.

5: Provide some more future perspectives.

>> Thanks for your suggestion. We have provided more future studies and application perspectives in the part of Conclusions.
